# Bioinformatics Approaches Applied to the Discovery of Antifungal Peptides

**DOI:** 10.3390/antibiotics12030566

**Published:** 2023-03-13

**Authors:** Carmen Rodríguez-Cerdeira, Alberto Molares-Vila, Carlos Daniel Sánchez-Cárdenas, Jimmy Steven Velásquez-Bámaca, Erick Martínez-Herrera

**Affiliations:** 1Efficiency, Quality, and Costs in Health Services Research Group (EFISALUD), Galicia Sur Health Research Institute (IISGS), Servizo Galego de Saúde-Universidade de Vigo (UVIGO), 36213 Vigo, Spain; 2Dermatology Department, Hospital do Vithas, 36206 Vigo, Spain; 3Fundación Vithas, Grupo Hospitalario Vithas, 28043 Madrid, Spain; 4Department of Health Sciences, University of Vigo, Campus of Vigo, As Lagoas, 36310 Vigo, Spain; 5European Women’s Dermatologic and Venereologic Society (EWDVS), 36700 Tui, Spain; 6Mycology Task Force of the Ibero-Latin American College of Dermatology (CILAD), Buenos Aires C1093AAA, Argentina; 7Psychodermatology Task Force of the Ibero-Latin American College of Dermatology (CILAD), Buenos Aires C1093AAA, Argentina; 8Microbiology Department, University Clinical Hospital of Santiago de Compostela, Servicio Galego de Saúde (SERGAS), 15706 Santiago de Compostela, Spain; 9Research Methods Group (RESMET), Health Research Institute of Santiago de Compostela (IDIS), Servicio Galego de Saúde (SERGAS)-Universidade de Santiago de Compostela (USC), 15706 Santiago de Compostela, Spain; 10Dermatology Department, Centro Médico Nacional La Raza, Paseo de Las Jacarandas S/N, La Raza, Azcapotzalco, Ciudad de Mexico 02990, Mexico; 11Dermatology Department Hospital General San Juan de Dios, 1ra Avenida “A” 10-50, Zona 1, Ciudad de Guatemala 01001, Guatemala; 12Sección de Estudios de Posgrado e Investigación, Escuela Superior de Medicina, Instituto Politécnico Nacional, Plan de San Luis y Díaz Mirón s/n, Col. Casco de Santo Tomas, Alcaldía Miguel Hidalgo, Ciudad de Mexico 11340, Mexico

**Keywords:** bioinformatics tools, antifungal peptides, databases, computational tools

## Abstract

Antifungal peptides (AFPs) comprise a group of substances with a broad spectrum of activities and complex action mechanisms. They develop in nature via an evolutionary process resulting from the interactions between hosts and pathogens. The AFP database is experimentally verified and curated from research articles, patents, and public databases. In this review, we compile information about the primary databases and bioinformatics tools that have been used in the discovery of AFPs during the last 15 years. We focus on the classification and prediction of AFPs using different physicochemical properties, such as polarity, hydrophobicity, hydrophilicity, mass, acidic, basic, and isoelectric indices, and other structural properties. Another method for discovering AFPs is the implementation of a peptidomic approach and bioinformatics filtering, which gave rise to a new family of peptides that exhibit a broad spectrum of antimicrobial activity against *Candida albicans* with low hemolytic effects. The application of machine intelligence in the sphere of biological sciences has led to the development of automated tools. The progress made in this area has also paved the way for producing new drugs more quickly and effectively. However, we also identified that further advancements are still needed to complete the AFP libraries.

## 1. Introduction

Fungal infections in humans are constantly evolving with multiple resistance mechanisms developing in fungi [1]. Therefore, discovering effective treatments against these pathogens is challenging [2,3]. Additionally, the inadequacy of both antifungal therapies and their negative side effects on patients has resulted in new investigations focused on finding new antifungal peptides (AFPs) [4] as possible alternatives for the treatment of fungal infections. AFPs are a group of natural peptides that are obtained from different species of bacteria, archaea, and eukarya. These are isolated from natural sources and are fundamental to their own native immune systems [5]. Importantly, AFPs exhibit low toxicity in mammals, making them eligible for use as peptide-based drugs.

The mechanisms of action of AFPs have been widely described in the literature, particularly those obtained from microorganisms [6] and plant-derived antifungal peptides (phytoAFP) [6].

However, the discovery of AFPs is hampered by challenges related to their production, as special humidity and temperature conditions are required. Therefore, AFP production requires a significant effort and a large amount of time [7]. Over the past few decades, numerous studies have investigated both natural and synthetic AFPs (Figure 1 and Figure 2). In particular, different AFPs vary according to their mechanisms of action in preventing biofilm formation, especially those related to *Candida* spp. [8] (Figure 3).

In addition, relevant documentation is available in many databases. Currently, various bioinformatics tools allow us to identify, check the dimensions of, and predict possible target sites of peptides in relation to protein–peptide complexes and the conformation of peptides within binding sites, which leads us to the possibility of designing antifungal drugs through the use of computational approaches, including artificial intelligence [9].

With computational methods, significant progress has been made, and substantial amounts of AFP data are now obtainable from publicly accessible data banks. Hence, patterns can be extracted to design new models for the prediction of AFPs [10].

The aim of this article is to offer an overview of the progress made in computational resources, bioinformatics approaches, and databases that assist in the prediction, design, and identification of new active AFPs for application to different organisms, including humans.

## 2. Results and Discussions

One of the main AFP reference databases, the antimicrobial peptide database (APD), was created in 2003. Five kingdoms (bacteria, protists, fungi, plants, and animals) of antimicrobial peptides (AMPs) are registered in this database. The APD content was curated from the literature via PubMed, PDB, and Swiss-Prot. Another resource (https://aps.unmc.edu/; accessed on 16 May 2022) provides useful information on the peptide-discovery process [11]. In addition, the construction of this curated database also enables the development of different strategies for peptide design. Based on the database, one research group has tested peptide screening [12,13] and database-guided design [14,15,16].

PlantAFP is a specific database that was recently created [17]. It is a repository of manually curated and verified plant-origin antifungal peptides gathered from scientific publications, patents, and other public databases. Initially, the database was conceived as a collection of natural-source drugs for plant health and focused on the discovery of new products in crop protection. However, it can also be applied to other organisms (including humans), since some AFPs are well-known infectious agents recognized to act against, e.g., *Aspergillus niger*, *Candida albicans*, etc. This database provides several tools, such as BLAST, peptide, or SMILE searches, peptide mapping, and additional information regarding the number of peptides, their family, and their presence in plant species (http://bioinformatics.cimap.res.in/sharma/PlantAFP/; accessed on 16 May 2022) [17].

Computational software for peptide sequence prediction is an AMP-identifying approach that is currently under development [18]. The software uses tools that scan the databases and correlate the curated AMP features with the amino acid sequences. The in silico identification of known AMP characteristics in new molecules helps shorten the annotation process and results in databases being more comprehensive [19]. Another bioinformatics algorithm applied to the APD database is the polarity index method [20], in which polarity is analyzed during the initial, intermediate, and final stages of a peptide and as much information about the peptide as possible is displayed, including its amino acid linear sequence, toxic action, etc. Polanco et al. [21] applied this method to identify AFPs with an efficiency of 70% true positives.

Mousavizadegan et al. [22] developed a computational approach for AFP classification and prediction based on a combination of Chou’s pseudo amino acid composition (PseAAC) [23] and the support vector machine classifier [9,24,25]. Firstly, AFPs were gathered from the APD, the Collection of Antimicrobial Peptides (CAMP) server (http://www.camp.bicnirrh.res.in; accessed on 16 May 2022) [26], the PhytAMP database (http://phytamp.hammamilab.org/entrieslist.php; accessed on 18 May 2022) [27], and a PubMed search. As a training set for data processing and classification, an independent dataset (100 proven AFPs + 100 proven non-AFPs) was created from the primary datasets using a combination of physicochemical parameters such as hydrophobicity, hydrophilicity, mass, acidic, basic, and isoelectric indexes [22]. A better performance of the classifier was reached by selecting a subset of the best features via the backward elimination search method available on the RapidMiner platform (2018 version) [28,29]. After exhaustive validation of the developed model, 22 derived peptides from the P24 protein of the HIV-1 virus were predicted [22]. In vitro assays on the three peptides with the highest prediction score showed a precision of 94.76% in the prediction of true positives and true negatives with the PseAAC + support vector machine (SVM) method [22].

Amaral et al. created in-house software for the identification of AMP sequences in fungus transcriptomes (*Paracoccidioides brasiliensis)* and human genomes [30]. Four AMPs were identified: P2 + P3 in *P. brasiliensis* and P1 + P4 in *H. sapiens*. The P1 and P2 showed antifungal properties against *Candida albicans* (tested in vitro). The use of in silico tools, such as Modeller v9.8 (https://salilab.org/modeller/; accessed on 16 May 2022), the I-Tasser server (https://zhanggroup.org/I-TASSER/; accessed on 16 May 2022) [31], PROCHECK (https://www.ebi.ac.uk/thornton-srv/software/PROCHECK/; accessed on 16 May 2022) [32], and 3DSS (http://cluster.physics.iisc.ernet.in/3dss/; accessed on 16 May 2022) [33], helped to establish the relation between structure and function.

In another study, cruzioseptins (a new family of AMPs from *Cruziohyla calcarifer*) were discovered through a peptidomic and bioinformatics approach [34]. The MEGA v6.0 (https://www.megasoftware.net; accessed on 18 May 2022) and BLAST tools (https://blast.ncbi.nlm.nih.gov/Blast.cgi; accessed on 18 May 2022) were applied to the nucleotide sequences [35,36]. Additionally, SignalP 4.1 server (https://services.healthtech.dtu.dk/service.php?SignalP-4.1; accessed on 16 May 2022), peptide-mass calculator v3.2 (http://rna.rega.kuleuven.be/masspec/pepcalc.htm; accessed on 18 May 2022), GOR4 (https://npsa-prabi.ibcp.fr/cgi-bin/npsa_automat.pl?page=npsa_gor4.html; accessed on 23 May 2022), HeliQuest computational parameters (https://heliquest.ipmc.cnrs.fr; accessed on 23 May 2022), and the peptide property calculator, Bachem (https://www.bachem.com/knowledge-center/peptide-calculator/; accessed on 25 May 2022), were used for signal peptide prediction [37], theoretical peptide mass calculations [38], secondary structure prediction [39], and physicochemical property calculations [40,41], respectively. The software applications used were versions from 2016. These AMPs showed antimicrobial activity against *S. aureus*, *E. coli*, and *C. albicans* with low hemolytic effects [34].

In a review, Neelabh et al. [42] presented a comprehensive summary of the known AFPs that are naturally produced by prokaryotes and eukaryotes, and discovered by the use of bioinformatics tools. The reported AFPs cover a wide spectrum of mammalian defensins, protegrins, tritrpticins, histatins, lactoferricins, and AFPs—found in birds, amphibians, insects, fungi, and bacteria—in addition to their synthetic analogs [42]. These include pexiganan, omiganan, echinocandins, and novexatin. In addition, bioinformatics tools applied to AFP discovery include multiple sequence alignment tools (T-Coffee; https://tcoffee.crg.eu/apps/tcoffee/do:regular; accessed on 25 May 2022 [43]) and structure visualization tools (UCSF Chimera; http://www.cgl.ucsf.edu/chimera/index.html; accessed on 25 May 2022 [44]), also in versions from 2016 [42].

In another very recent study, models using machine learning techniques were developed to differentiate AFPs from natural and other AMPs [45]. The working team extracted exclusive AFPs from the DRAMP database (Data Repository of Antimicrobial Peptides) [46] to create training and validation datasets that are used to evaluate the performances of all the tested models: SVM, random forest, naïve Bayes, etc. The best-performing model was a binary-profile-based model that can be used to differentiate with good accuracy between sequences that are very similar, but perform different activities. The three top-scoring models were selected to develop a class-specific prediction web server for AFPs called “Antifp”. This web server also provides a tool to design new analogs with enhanced antifungal properties. It is freely accessible at http://webs.iiitd.edu.in/raghava/antifp (accessed on 26 May 2022). Furthermore, a standalone version was developed for installation on Linux, Mac, or Windows 64-bit operating systems [45].

Finally, a similar study was conducted on the development of an SVM-based model for the design and prediction of therapeutic phytoAFP [6]. The two best models were selected by using functions such as mono-, di-, and tripeptide compositions. On the other hand, using binary, hybrid, and physiochemical properties: a TPC-based monopeptide-composition model (accuracy = 94.4% with a Matthews correlation coefficient, or MCC, of 0.89) and a model based on dipeptides (accuracy = 94.28% with MCC = 0.89). The datasets were constructed with manually curated peptides extracted from the PlantAFP and UniProt databases. In addition, a PhytoAFP web-based prediction server was developed for predicting and designing therapeutic antifungal peptides in a user-friendly manner. This server also provides a module for generating new mutants or analogs and predicting the activity and physicochemical properties of AFPs. It can be accessed at http://bioinformatics.cimap.res.in/sharma/PhytoAFP (accessed on 26 May 2022) [6].

Once AFPs have been discovered, other bioinformatics tools are used for the analysis of their mechanisms of action. For example, Amaral et al. used the PEP-FOLD server (https://bioserv.rpbs.univ-paris-diderot.fr/services/PEP-FOLD3/; accessed on 16 May 2022) [47], the PROPKA 3.1 web server tool (https://github.com/schrodinger/propka-3.1; accessed on 26 May 2022) [48], the Biovia Discovery Studio suite, version 3.1 [49], Autodock [50], GROMACS (version 2018.4) [51], and VMD software (versions from 2021) [52] for in silico analysis and performed microscopic assays of Mo-CBP3-PepIII, an anticandidal peptide. These investigations demonstrated that Mo-CBP3-PepIII affects the cell wall and membrane of *C. albicans.* Furthermore, it was found that multiple mechanisms of action are potentially involved in similar actions of antifungal agents belonging to the class of echinocandins [53].

Table 1 lists the most relevant results found with the databases previously described, as well as bioinformatics approaches, for the discovery of new AFPs.

## 3. Materials and Methods

This systematic review is reported in accordance with the PRISMA guidelines (https://prisma-statement.org/Default.aspx; accessed on 12 April 2022) [54].

The PubMed and Embase databases were searched in April 2022 for articles on AFPs from human subjects, amphibians, birds, fish, and plants. The following search terms were used: (Antifungal peptides) “[All Fields]” OR (Fungi) OR (Database screening) OR (Databases) OR (Bioinformatic) OR (Machine learning methods) OR (Antifungal peptides resistance) OR (Membrane lipids) AND (Human immune system) AND (New therapies) OR (Production) “[All Fields]” OR (Therapeutic antifungal agents) OR (Defensins) OR (Protegrin) OR (Amphiphilic) OR (Novel nanomaterials).

Revisions and guidelines were included, along with non-English articles. Primary outcomes included the source, structure, and toxic effects of AFPs.

Some cases of unavailable data (NA) were found. Data extracted from bibliographic searches are illustrated in the PRISMA flow diagram (Figure 4).

## 4. Conclusions

Bioinformatics tools are used for antifungal peptide discovery as part of a global strategy to discover AMPs from a variety of organisms covering the six kingdoms (Animalia, Plantae, Fungi, Protista, Archaea, and Bacteria). For this purpose, comprehensive databases serve as reference sites to find curated information about known AFPs. Furthermore, with the help of statistical classification and prediction methods, the first steps have been taken toward the prediction of new molecules with specific sequence characteristics and physicochemical properties. However, further research is needed to optimize and validate the best methods for establishing an international, standardized methodology.

## Figures and Tables

**Figure 1 antibiotics-12-00566-f001:**
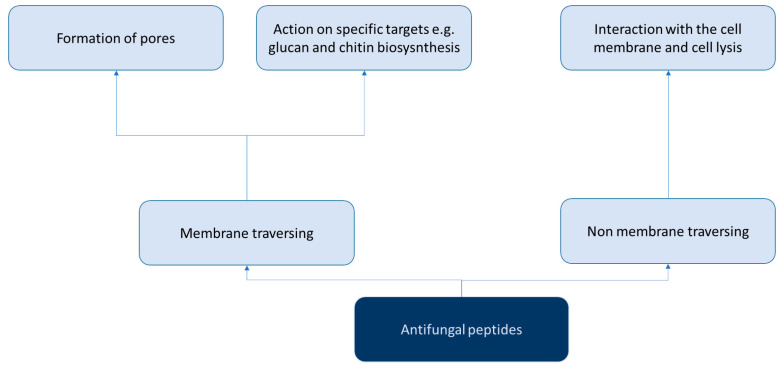
Scheme of the classification of antifungal peptides according to the mechanism of action in general.

**Figure 2 antibiotics-12-00566-f002:**
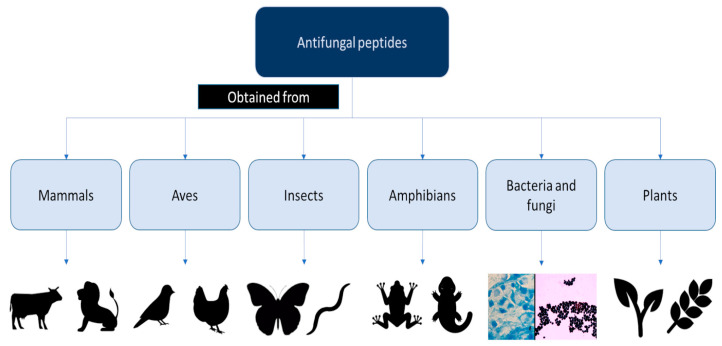
Drawing of the different antifungal peptide origins.

**Figure 3 antibiotics-12-00566-f003:**
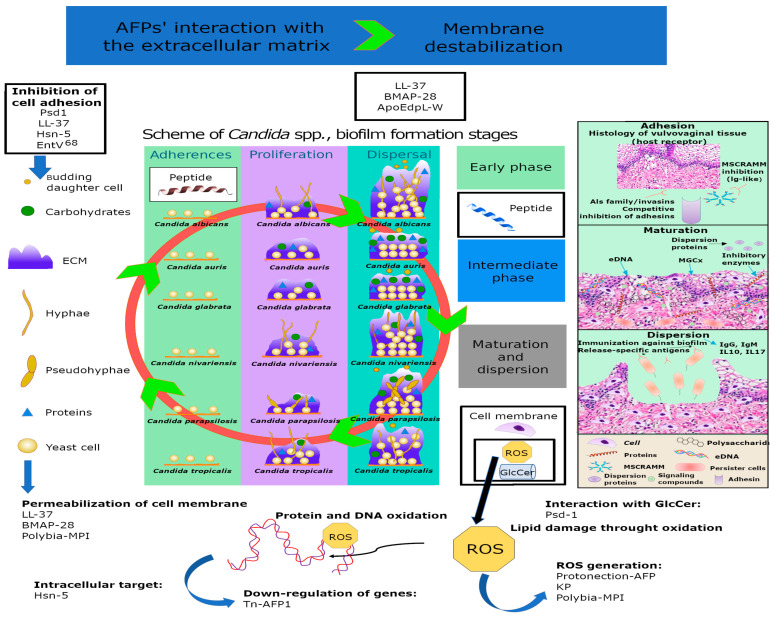
Mechanisms of action of AFPs that trigger antibiofilm activities against *Candida* spp. and cause biofilm destabilization. AFPs: antifungal peptides; ECM: extracellular matrix; MSCRAMM: microbial surface components recognizing adhesive matrix molecules; MGCx: extracellular matrix mannan-glucan complex; eDNA: external DNA. ROS: reactive oxygen species; KP: Killer peptide. (EntV)^68^: *Enterococcus faecalis* bacteriocin; Hsn-5: 5-Hydroxytryptophan. Peptides with intracellular mechanisms that trigger antibiofilm activities. In the boxes are the names of the peptides involved in the different mechanisms of action: Psd1 (PDB: 1JKZ); LL-37 (PDB: 2K60); BMAP-28 (PDB: 2NDC); ApoEdpL-W: Peptide derived from human ApoE apolipoprotein. Tn-AFP1: antifungal peptide.

**Figure 4 antibiotics-12-00566-f004:**
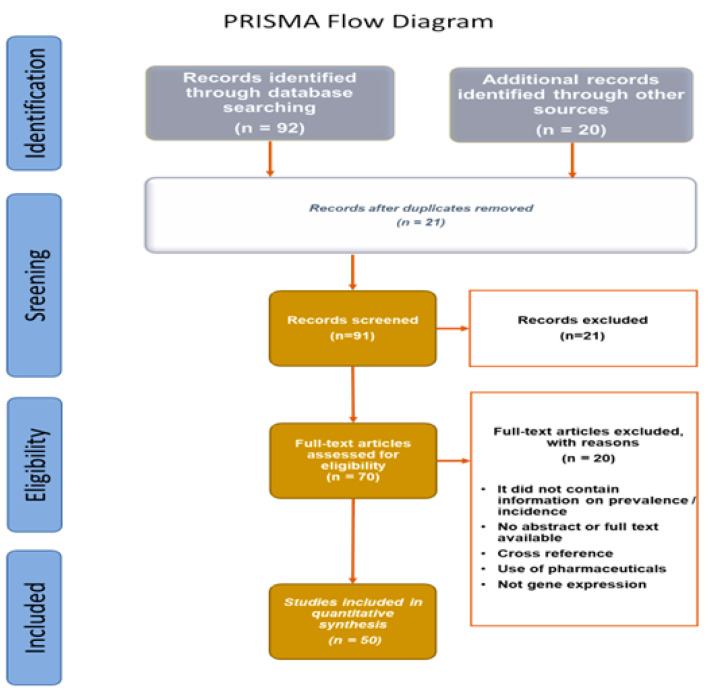
PRISMA flow diagram of data extracted from bibliographic searches.

**Table 1 antibiotics-12-00566-t001:** Computer tools applied to the identification, discovery, or molecular dynamics of antifungal peptides (AFPs).

Computational Resources	Utility	Discussion	References
APD	Reference site for AFPs	Complete database	[11]
PlantAFP	Repository for plant-derived AFPs	Complete database	[17]
Polarity Index	Identification of AFPs	More suitable for bacteria than fungi; efficiency > 90%	[20]
In-house method	AFP classification and prediction	Positive validation assays performed in vitro for peptides with high antifungal prediction score (>0.95)	[22]
In-house method	Identification of AFP sequences from *P. brasiliensis* and *H. sapiens*	Four highest-scoring peptides were selected in silico and checked in vitro; two peptides had weak antifungal activity against *Candida albicans*	[30]
In-house method	Identification of AFP sequences from *C. calcarifer*	Antimicrobial activity against *C. albicans* found in three synthetic peptides	[33]
In-house method	Discovery of AFPs produced naturally by prokaryotes and eukaryotes	Review of some AFPs produced in mammals, birds, insects, amphibians, and microbes based on their structural characterization	[42]
Antifp	Class-specific prediction web server for AFPs	Differentiates with good accuracy between sequences that are very similar in identity but possess different activities	[45]
PhytoAFP	Web prediction server	Prediction and design of plant-derived antifungal peptides	[6]
In-house method	Mechanism-of-action analysis of antifungal agents	In silico assays (molecular docking and dynamics simulations) indicated that cell wall and membrane of *C. albicans* are targeted by Mo-CBP3-PepIII	[53]

AFPs: antifungal peptides; APD: antimicrobial peptide database.

## Data Availability

Not applicable.

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
