# Peer review of "Bioinformatics Approaches Applied to the Discovery of Antifungal Peptides"

_antibiotics, 2023, doi:10.3390/antibiotics12030566_

Round 1

Reviewer 1 Report

This manuscript review is too general, only mentioning some computational tools that helped those authors in their research and findings. It would be more convenient if sequence, structure and biological activities had been exhibited in figures, tables, and alike. This is very important for example to allow comparison or establish relationship between organisms/protein target and structure. I suggest you to improve the content with figures, tables, to help readers understand the relevant findings and allow the comparison between structures and biological in/activities. In addition, the PRISMA statement is a recommendation of how to follow a systematic way to write a review. It is important to notice that as a guideline it is very useful, but authors should also provide the deep consistence of the manuscript according to the journal impact and recommendations. For example, see the review available at https://www.mdpi.com/2079-6382/12/1/42. In this review those authors comment structure and the mechanism of action of some peptides. Moreover, with more and deeper information the review will bring increased visibility and help new authors in their researches.

Here I point some minor corrections to be made:

Line 50: It looks like this sentence lacks a connector to tie it to the text;

Line 53: “…makingthem…”;

Line 54: dot in the end of the sentence;

Line 63: the idea of this paragraph and the next one seems to be the same; then, they could be kept together.

Line 80: double dots.

Line 82: what does NA mean?

Figure 1: letters of the flow diagram are too small for readers.

Line 96: English correction (“…is been…”)

Line 99: word correction

Author Response

First of all, we want to thank the reviewers for the time they spent on our review. Your contributions were of great help to us.

Reviewer 2 Report

-    Line 28: replace "broad spectrum" with "a broad spectrum".

-  Line 35: it is correct to rewrite the sentence with “another method for discovering”.

-        Lines 50 and 53-54: sentences seem suspended and references are missing.

-   Lines 51 and 52: it is said that AFPs are fundamental peptides for the immune system but it is not specified to which organisms are referred.

-        Line 60: specify which type of characterization was made.

-        Line 80: at the end of the sentence there are two full stops.

-        Line 82: specify what is meant by the abbreviation in brackets.

-        Figure 1: it would be better to redo the diagram in a better graphic.

-        In “Materials and Methods” please add the link to PRISMA.

-        The link at line 91 doesn’t work: it’s better to control it.

-        Line 102: rewrite et al. in italics, check others along the text.

-        In some points of the paragraph "Results and Discussion" it is necessary at the end of the sentence to add or repeat the references (lines 111, 115, 117, 132, 137, 140).

- In Results and Discussion when naming PubMed, PDB, Swiss-Prot, CAMP, I-Tasser and all mentioned servers it is preferable to add the reference link for each one.

-        I recommend to mention also the bioinformatic tool Antifungal peptide Prediction (Antifp) present at the link https://webs.iiitd.edu.in/raghava/antifp/predict3.php and well described in DOI:10.2174/1381612828666220817163339, that I suggest to mention.

-        As the paper is a review, I suggest not to call sections as “Material and methods” and “Results and discussion”.

-        Please expand the conclusion section.

Author Response

(The authors gave the same response as above.)

Round 2

Reviewer 1 Report

Review consideration and suggestions of the version 3:

Line 2: The title would be more adequate to the content if changes from “Bioinformatics tools…” to “Bioinformatics approaches…”. Usually the term “tools” is employed to specific computational programs, and some “tools” in the manuscript are databases.

Line 60:  Here is a more concise text “However, the discovery of AFPs is hampered by challenges related to their production, as special humidity and temperature conditions are required.”

Line 69: “... which leads us to the possibility of designing antifungal drugs through the use of artificial intelligence.” It is better not to consider only artificial intelligence but computational tools in general. As a suggestion, change the words “… the use of artificial intelligence” to “… the use of computational approaches, including artificial intelligence” which is broader and considers also findings reported which did not use artificial intelligence. Artificial intelligence is only part of those computational approaches. It is also in agreement with the suggestion for the tittle (“Bioinformatics approaches…” instead of “Bioinformatics tools…”)

Line 77 and 78 (Figure 1): “Scheme of the …action generated.” It would be better to change to “… action in general.”

Figure 2: “Aves” to “Birds”

Figure 3: It is necessary corrections like “Budding dauther cell” to “Budding daughter cell”. What does “Battacion” means? Is it a peptide? Some explanation of about the mechanism would be necessary. Only showing the figures without explaining them will not clarify the mechanisms. Also, the words in figures are too small. Even checking figures on a computer, it was not possible to read the words clearly. It would be great to use higher fonts with higher figures. Also, the legend of the figure did not include any explanation about the mechanisms. At least this would be in the text to emphasize every picture. It is necessary to explain the contents of this figure for readers to understand better what is being showed inside it. Please, verify the entire Figure 3. Did the pictures from Figure 3 come from another source or they are original?

Line 92: It would be better to change the term “bioinformatics tools” to “bioinformatics approaches”. It includes not only computational programs but also databases.

Line 92 to line 98. The aim is not clear. First, it states that the involves “…bioinformatics tools…” (suggested to change for "approaches") “…that assist in the ... active AFPs for application to different organisms, including humans.” After, it suggests another meaning when mentioning that “this systematic review was carried out in order to summarize the information on new AFPs in the management and treatment of mycoses…”, although “mycoses” word was not mentioned in the search method but “antifungal” in general. Then, a doubt arises if the aim is discovering AFPs in general with the aid of computational approaches or only those AFPs that show activity against mycoses diseases. Here, it is necessary to establish a concise explanation of the aim.

Figure 4: Words correction like “Sreening” to “Screening”.

Line 219: This paragraph looks like a part of the text from last one.

Line 223: It states the most relevant results, but in a context that it looks relative to the last paragraph only. It needs connection when describing to include all results previously mentioned. For example, “Table 1 lists the most relevant results found with the AFPs databases previously described and bioinformatics approaches…”. Also, the column “Computer Tool” does not apply to databases, as a database is not a computer tool. Then, I suggest modifying this column title to a more appropriate one.

Author Response

mments and Suggestions for Authors

Review consideration and suggestions of the version 3:

Line 2: The title would be more adequate to the content if changes from “Bioinformatics tools…” to “Bioinformatics approaches…”. Usually the term “tools” is employed to specific computational programs, and some “tools” in the manuscript are databases.

WE DID

Line 60:  Here is a more concise text “However, the discovery of AFPs is hampered by challenges related to their production, as special humidity and temperature conditions are required.”

WE DID

Line 69: “... which leads us to the possibility of designing antifungal drugs through the use of artificial intelligence.” It is better not to consider only artificial intelligence but computational tools in general. As a suggestion, change the words “… the use of artificial intelligence” to “… the use of computational approaches, including artificial intelligence” which is broader and considers also findings reported which did not use artificial intelligence. Artificial intelligence is only part of those computational approaches. It is also in agreement with the suggestion for the tittle (“Bioinformatics approaches…” instead of “Bioinformatics tools…”)

WE DID

Line 77 and 78 (Figure 1): “Scheme of the …action generated.” It would be better to change to “… action in general.”

WE DID

Figure 2: “Aves” to “Birds”

WE DID

Figure 3: It is necessary corrections like “Budding dauther cell” to “Budding daughter cell”. What does “Battacion” means? Is it a peptide? Some explanation of about the mechanism would be necessary. Only showing the figures without explaining them will not clarify the mechanisms. Also, the words in figures are too small. Even checking figures on a computer, it was not possible to read the words clearly. It would be great to use higher fonts with higher figures. Also, the legend of the figure did not include any explanation about the mechanisms. At least this would be in the text to emphasize every picture. It is necessary to explain the contents of this figure for readers to understand better what is being showed inside it. Please, verify the entire Figure 3. Did the pictures from Figure 3 come from another source or they are original?

WE DID

Line 92: It would be better to change the term “bioinformatics tools” to “bioinformatics approaches”. It includes not only computational programs but also databases.

WE DID

Line 92 to line 98. The aim is not clear. First, it states that the involves “…bioinformatics tools…” (suggested to change for "approaches") “…that assist in the ... active AFPs for application to different organisms, including humans.” After, it suggests another meaning when mentioning that “this systematic review was carried out in order to summarize the information on new AFPs in the management and treatment of mycoses…”, although “mycoses” word was not mentioned in the search method but “antifungal” in general. Then, a doubt arises if the aim is discovering AFPs in general with the aid of computational approaches or only those AFPs that show activity against mycoses diseases. Here, it is necessary to establish a concise explanation of the aim.

WE DID

Figure 4: Words correction like “Sreening” to “Screening”.

WE DID

Line 219: This paragraph looks like a part of the text from last one.

WE DID

Line 223: It states the most relevant results, but in a context that it looks relative to the last paragraph only. It needs connection when describing to include all results previously mentioned. For example, “Table 1 lists the most relevant results found with the AFPs databases previously described and bioinformatics approaches…”. Also, the column “Computer Tool” does not apply to databases, as a database is not a computer tool. Then, I suggest modifying this column title to a more appropriate one.

WE DID

Reviewer 2 Report

The authors made all the corrections and suggestions requested.

Author Response

No new suggestions
Thank you so much